# Two-Person Interaction Augmentation with Skeleton Priors

## Abstract

*Close and continuous interaction with rich contacts is a crucial aspect of human activities (e.g. hugging, dancing) and of interest in many domains like activity recognition, motion prediction, character animation, etc. However, acquiring such skeletal motion is challenging. While direct motion capture is expensive and slow, motion editing/generation is also non-trivial, as complex contact patterns with topological and geometric constraints have to be retained. To this end, we propose a new deep learning method for two-body skeletal interaction motion augmentation, which can generate variations of contact-rich interactions with varying body sizes and proportions while retaining the key geometric/topological relations between two bodies. Our system can learn effectively from a relatively small amount of data and generalize to drastically different skeleton sizes. Through exhaustive evaluation and comparison, we show it can generate high-quality motions, has strong generalizability and outperforms traditional optimization-based methods and alternative deep learning solutions.*

## 1. Introduction

Skeletal motion is a crucial data modality in many applications, such as human activity recognition, motion analysis, security and computer graphics [8, 29, 42, 50, 51, 53]. However, capturing high-quality skeletal motions often requires expensive hardware, professional actors, costly post-processing and laborious trial-and-error processes [34]. Affordable devices such as RGB-D cameras can reduce the cost but usually provide data with jittering and tracking errors [38]. As a result, the majority of available skeletal data is based on single-person [31] or multiple people with short, simple and almost-no-contact interactions [38]. Datasets with close and continuous interactions [12] are rare, limiting the research of motion generation [54], prediction [12], classification [42] within such motions.

One way to tackle the challenge is to carefully capture the motion of actors and retarget it onto different skele-

tons [13]. With a single skeleton, the problem can be formulated as optimizations with respect to keeping key geometric and dynamic constraints [3, 43]. However, this process quickly becomes intractable with the increase of constraints such as foot contact and hand-environment contact, let alone retargeting two people with close and continuous interactions like wrestling and dancing, where inter-character geometric/topological constraints need to be retained [14, 30]. Consequently, multiple runs of complex optimization with careful hand-tuning of objective function weights are needed [15, 16] for a single motion, which is prohibitively slow and therefore can only be used to generate small amounts of data.

Meanwhile, data-driven approaches for single body retargeting [4], despite being successful, cannot be directly extended to two-character interaction. Methodologically, these methods do not model inter-character geometric constraints, which is key to the semantics of interactions [16]. From the data point of view, these approaches, especially those using deep learning [2, 48], require a large amount of data, which is largely absent for two-character interaction. Existing two-character interaction datasets are for action recognition [7, 37] and low-quality, or only consist of a small amount of data with limited variations in body sizes [12], hardly covering the distribution of possible body variations. Considering the high cost of obtaining interaction data, a method that can learn effectively from limited data and generate interactions with diversified body variations is highly desirable.

We propose a novel lightweight framework for two-character skeletal interaction augmentation, easing the need to capture a large amount of data. Our key insight is the joint relations evolving in time (e.g. relative positions, velocities, etc.) can fully describe an interaction, e.g. hugging always involves wrapping one's arms around the other's body. These relations change when the body size changes, but the *distribution* of them should stay similar in the sense that one's arms should still wrap around the other, such that the hand-to-body distance is always smaller than e.g. the foot-to-body distance. Meanwhile, this distribution should be very different from other types of interactions

e.g. wrestling. Therefore, to generate motions from different skeleton sizes, the key is being able to predict the joint relation distributions based on a given skeleton.

To this end, we propose a conditional motion generation approach, where the generated motions are conditioned on the joint relation distribution which is further conditioned on a skeleton prior, allowing a skeleton change to propagate through the joint relation distribution and finally influence the final motion. We start by modeling the joint probability of two-body motions and proposing a novel factorization to decompose it into three distributions. The three distributions are realized as neural networks, which together form an end-to-end model that conditions two-body motions on one person's body size. Further, to address the data scarcity challenge, we capture new two-body data and employ an existing optimization-based method for initial data augmentation. After training our model on the data, it can be employed for further motion data augmentation for many downstream tasks.

We evaluate our method in multiple tasks. Since there is no similar method for baselines, we compare our method with adapted baselines and optimization-based approaches, demonstrating that our method is accurate in generating desired motions, can generate diversified interactions while respecting interaction constraints, is much faster for inference and generalizes to large skeletal changes than optimization-based methods. In addition, our model benefits downstream tasks including motion prediction and activity recognition. Formally, our contributions include:

1. A new factorization of two-character interactions that allows for effective modelling of interaction features.
2. a new deep learning method for interaction retargeting/generation to the best of our knowledge, which learns and generalizes effectively from a small number of training samples.
3. A new dataset augmented from single interaction examples, containing interactions with different body sizes and proportions.

## 2. Related Work

### 2.1. Deep Learning for Skeletal Motion

Neural networks have been successful in modeling skeletal motions. Convolutional neural networks can learn latent representations for denoising and synthesis [18]. Recurrent neural networks improve the robustness and enable long horizon synthesis [5, 52]. Graph neural networks capture the joint relations [27]. Generative flows combine the style and content in the latent space [57]. Transformers co-embed human motion and body parameters into a latent representation [35]. Diffusion models provide a larger capacity and are less prone to mode collapse in generation [47, 64]. But all the above research is on a single body. While there is some research in modeling human-environment interactions [20, 61], two-body interactions are more complex. Very recent research shows successful synthesis of two interacting characters, but their focus is either on single character control [24, 41], or fix one while generating the other [10, 28]. None of them models interactions, especially under varying body sizes and proportions. To our best knowledge, there is no deep-learning method for complex two-character interactions.

### 2.2. Motion Retargeting

Motion retargeting adapts a character's motion to another of a different size while maintaining the motion semantics. Early research employs space-time optimization based on contact [9], purposefully-designed inverse kinematics solver for different morphologies [13], data-driven reconstruction of poses based on end-effectors [4, 40], or physical filters [43] and physical-based solvers [3] considering dynamics constraints. Recently, deep learning has achieved great success, e.g. recurrent neural networks with contact modeling [49], skeleton-aware operators without explicitly pairing the source and target motions [2], and variational autoencoders for motion features preservation during retargeting [48]. Beyond skeletal motions, the skeleton structure is also effective in video based retargeting [60]. Fast deep learning methods are pursued for real-time robotic control [63]. Unlike previous research, we propose a novel deep learning architecture for motion retargeting/generation of two-character interactions, which are intrinsically more complex than single-character retargeting.

### 2.3. Interaction

Interaction retargeting involving more than one person is more challenging than single-body retargeting, due to their complex motion constraints [22] such as topological constraints [14], but these constraints involve heavy manual designs. As a more general solution, InteractionMesh [16] uses dense mesh structures to represent the spatial relations between two characters and minimizes the mesh change during retargeting [17] and synthesis of character-environment interactions [15]. As it may result in unnatural movements when the skeleton is significantly different from the original one, a prioritization strategy on local relations is proposed [32]. Nevertheless, optimisation-based methods require careful design of constraints, and incur large run-time costs.

Recently, there is a surge of deep learning methods on interactions, including human-object interaction [21, 36, 58], motion generation as reaction [6], from texts [44] and by reinforcement learning [65]. Interaction has also been investigated in motion forecasting [33, 46, 59]. Among these papers, the closest work is interaction motion generation but existing work either cannot deal with skeletons of different

sizes or does not focus on continuous and close interactions. To our best knowledge, there is no deep learning method for interaction modeling as proposed in this research.

Another key bottleneck of two-character interaction retargeting/generation is the lack of data. Existing datasets focus on action recognition [7, 37, 62] with simple interactions. While some datasets with complex interactions are available [39], they include limited variations of body sizes/proportions and have a limited amount of data. In this research, we present a new dataset and a method that learns efficiently from small amounts of data.

## 3. Methodology

We denote a motion with $T$ frames as $q = \{q^0, \ldots, q^T\}^{\mathbf{T}} \in \mathbb{R}^{T \times N \times 3}$ where $q^t$ is the $t^{\text{th}}$ frame, and each frame $q^t = \{p_0^t, \ldots, p_N^t\}$ consists of $N$ joints and $p_j$ is the $j^{\text{th}}$ joint position. An interaction motion of two characters $A$ and $B$ is represented by $\{q_A, q_B\}$. For a specific interaction, different body sizes and proportions should not change the semantics, e.g. one character always having its arms around the other in hugging. These invariant semantics are often captured by topological/geometric features [14]. Therefore, a skeletal change in $B$ should cause changes in both $q_A$ and $q_B$ to retain the semantics. We represent a $B$ skeleton by its bone length vector $B_s \in \mathbb{R}^n$ where $n$ is the number of bones. The aim is to model the joint probability $p(B_s, q_A, q_B)$. We propose a simple yet effective model, shown in Fig. 1.

### 3.1. A New Factorization of Interaction Motions

Directly learning $p(B_s, q_A, q_B)$ would need large amounts of data containing different interactions with varying both lengths. Therefore, we first make it learnable on limited data by introducing a new factorization. First, we represent skeletons with different bone lengths as heterogeneously scaled versions of a *template* skeleton with a bone length scale vector $\hat{B} = \{1, ..., 1\} \in \mathbb{R}^n$, i.e. we treat the bone lengths of the template skeleton as scale 1. We abuse the notation and denote a skeleton variation by $B_s$, indicating how each bone is scaled with respect to $\hat{B}$.

Next, we represent motion data as deviations from some *template* motion $\{\hat{q}_A, \hat{q}_B\}$ with the template skeleton $\hat{B}$. A skeleton variation $B_s$ corresponds to a distribution of motions $\{q'_A, q'_B\}$, where not only the $B$ motion deviates from $\hat{q}_B$, the $A$ motion also deviates from $\hat{q}_A$ accordingly to maintain the interaction. So we can split data into template motions and others $\{q_A, q_B\} = \{\hat{q}_A, q'_A\} \bigcup \{\hat{q}_B, q'_B\}$, so that $p(B_s, q_A, q_B) = p(q'_A, q'_B, B_s, \hat{q}_A, \hat{q}_B)$. Given $\{\hat{q}_A, \hat{q}_B\}$, $p(q'_A, q'_B, B_s, \hat{q}_A, \hat{q}_B)$ is an easier distribution to learn than the original $p(B_s, q_A, q_B)$, as $\{\hat{q}_A, \hat{q}_B\}$ serves as an anchor motion with an anchor skeleton, so that all other motion variations can be described by offsets from the template motion, restricting $p(q'_A, q'_B, B_s, \hat{q}_A, \hat{q}_B)$ to only model the distribution of offsets from $\{\hat{q}_A, \hat{q}_B\}$.

There are many ways to factorize $p(q'_A, q'_B, B_s, \hat{q}_A, \hat{q}_B)$ theoretically. Our new factorization follows:

$$
\begin{aligned}
& p(q'_A, q'_B, B_s, \hat{q}_A, \hat{q}_B) \\
(i) = & \, p(q'_A | q'_B, B_s, \hat{q}_A, \hat{q}_B) p(q'_B, B_s, \hat{q}_A, \hat{q}_B) \\
(ii) = & \, p(q'_A | q'_B, \hat{q}_A) p(q'_B | B_s, \hat{q}_B) p(B_s, \hat{q}_A, \hat{q}_B) \\
(iii) = & \, p(q'_A | q'_B, \hat{q}_A) p(q'_B | B_s, \hat{q}_B) p(B_s) \quad (1)
\end{aligned}
$$

where (i) gives the conditional probability of $p(q'_A | q'_B, B_s, \hat{q}_A, \hat{q}_B)$, and its prior $p(q'_B, B_s, \hat{q}_A, \hat{q}_B)$. Further, $p(q'_B, B_s, \hat{q}_A, \hat{q}_B)$ can be factorized into $p(q'_B | B_s, \hat{q}_B) p(B_s, \hat{q}_A, \hat{q}_B)$ in (ii), assuming $q'_B$ does not depend on $\hat{q}_A$. Given the template motion $\{\hat{q}_A, \hat{q}_B\}$ and a changed skeleton $B_s$, $\{B_s, \hat{q}_A, \hat{q}_B\} \sim p(B_s, \hat{q}_A, \hat{q}_B)$, we can sample a new $q'_B \sim p(q'_B | B_s, \hat{q}_B)$ that satisfies the desired skeleton change, then further sample a new $q'_A \sim p(q'_A | q'_B, \hat{q}_A)$ that maintains the interaction with $q'_B$. Further, (iii) is obtained when $\{\hat{q}_A, \hat{q}_B\}$ is given.

The three distributions in Eq. (1) have explicit meanings. $p(B_s)$ is the *skeleton prior* which captures skeletal variations that are likely to be observed; $p(q'_B | B_s, \hat{q}_B)$ is for *motion retargeting*, i.e. modeling the distribution of possible $B$ motions w.r.t. $\hat{q}_B$, given a skeletal variation $B_s$; $p(q'_A | q'_B, \hat{q}_A)$ is for *motion adaptation*, i.e. modeling the possible $A$ motions w.r.t. $\hat{q}_A$, given a specific $B$ motion $q'_B$. Among many possible ways of factorization, our particular choice in Eq. (1) conforms to a plausible workflow where user input can be injected at multiple stages. The input can be a skeletal change $B_s$ to $p(q'_B | B_s, \hat{q}_B)$, or a keyframed new motion $q'_B$ to $p(q'_A | q'_B, \hat{q}_A)$. Alternatively, the $B_s$ can be drawn from $p(B_s)$ for unlimited motion generation.

To keep our model small, inspired by the recent research in human motions [25, 26], we learn a generative model by assuming $p(B_s)$, $p(q'_B | B_s, \hat{q}_B)$ and $p(q'_A | q'_B, \hat{q}_A)$ to have well-behaved latent distribution, e.g. Gaussian, shown in Fig. 1 Compared with other alternative networks such as flows and Transformers, our model is especially suitable since our data is limited. We introduce the general architecture and refer the readers to the supplementary material (SM) for details.

### 3.2. Network Architecture

In Fig. 1, MLP1 and MLP2 are a five-layer (16-32-64-128-256) fully-connected (FC) network, and a five-layer (256-128-64-32-dim($B_s$)) FC network, respectively. As $B_s$ is a simple n-dimensional vector with fixed structural information, i.e. each dimension representing the scale of a bone, simple MLPs work well in projecting $B_s$ into a latent space where it conforms to a Normal distribution.

Next, we choose two types of networks as key components of our model to learn motion dynamics and interactions. First, spatio-temporal Graph Convolution Networks

Figure 1. Overview of our model. The key components include Spatial-temporal Graph Convolution Networks (ST-GCN), Multi-layer perceptrons (MLP) and G-GRU networks. Details are in the supplementary material (SM).

(ST-GCN) extract features by conducting spatial and temporal convolution on graph data and have been proven effective in analyzing human motions [8, 51]. We use ST-GCNs as encoders to extract reliable features. The other network is a Recurrent Neural Network named Graph Gated Recurrent Unit or G-GRU [25]. G-GRU models time-series data by Gated Recurrent Unit on graph structures and have the ability to stably unroll into the future on predicting human motions [25]. We use it as decoders in our model. This choice is again for reducing the required amount of data for training, which would be much larger if other networks, e.g. ST-GCNs are used as decoders based on our experiments.

Instead of directly learning the distribution of $q'_B$, learning the distribution of the differences $\triangle q_B = q'_B - \hat{q}_B$ is easier [45, 52]: $p(q'_B|B_s, \hat{q}_B) = p(\triangle q_B|B_s)$, which is easier as it becomes learning the distribution of offsets from the template motion $\hat{q}_B$ and a skeleton variation $B_s$. We encode $\triangle q_B$ into a latent space then decode it back to the data space by:

$$z = \text{FC}(\text{Concat}(\text{ST-GCN1}(\triangle q_B, B_s), \hat{q}_B^0, \hat{q}_B^T)))$$
$$\triangle q'_B = \text{G-GRU1}(z, \hat{q}_B^0, \hat{q}_B^T, B_s))$$
$$\text{subject. to } z \sim \mathcal{N}(0, \mathbf{I}) \quad (2)$$

where in both the encoding and decoding processes, we also incorporate the first and last frame of the template motion $\hat{q}_B^0, \hat{q}_B^T$ because they help stabilize the dynamics based on our results. After decoding, we add the predicted $\triangle q'_B$ back to the template motion to get the new motion $q'_B = q_B + \triangle q'_B$.

Next, given a motion $q'_B$, character $A$ needs to adjust its motions to keep the interaction, leading to a distribution of possible $q'_A$. Similarly, we focus on learning $\triangle q_A = q'_A - \hat{q}_A$ by an autoencoder:

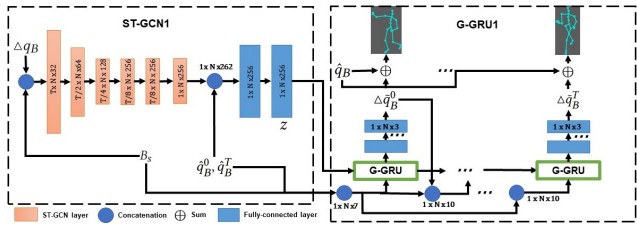

Figure 2. The architecture of ST-GCN1 and G-GRU1. More details are in the supplementary material.

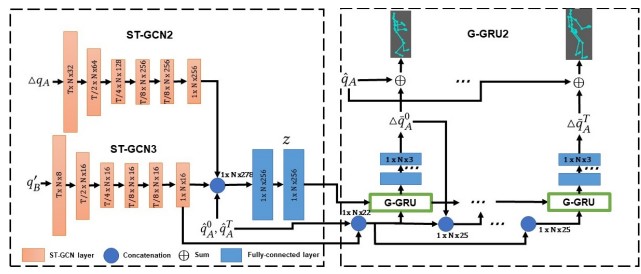

Figure 3. The architecture of ST-GCN2, ST-GCN3 and G-GRU2. More details are in the supplementary material.

$$z = \text{FC}(\text{Concat}(\text{ST-GCN2}(\triangle q_A), \hat{q}_A^0, \hat{q}_A^T, \text{ST-GCN3}(q'_B))$$
$$\triangle q'_A = \text{G-GRU2}(z, \hat{q}_A^0, \hat{q}_A^T)) \text{ subject to } z \sim \mathcal{N}(0, \mathbf{I}) \quad (3)$$

where after decoding we compute the new motion $q'_A = \hat{q}_A + \triangle q'_A$.

We give more detailed architectures of ST-GCN1 and G-GRU1 in Figure 2, and the detailed architectures of ST-GCN2, ST-GCN3 and G-GRU2 in Figure 3.

### 3.3. Loss functions

Training our model involves three loss terms for the three autoencoders:

$$\mathcal{L} = \mathcal{L}_{B_S} + \mathcal{L}_{B_M} + \mathcal{L}_{A_M}. \quad (4)$$

Minimizing $\mathcal{L}_{B_S}$ learns MLP1 and MLP2 to learn the distribution of possible skeleon variations $B_s$:

$$\mathcal{L}_{B_S} = \frac{1}{M}\sum ||B'_s - B_s||^2_2 + D_{KL}[z||\mathcal{N}(0,\mathbf{I})], \quad (5)$$

where $z$ is the output of MLP1, $B'_s$ is the output of MLP2, $B_s$ is the ground-truth skeleton variation and $D_{KL}$ is the KL-divergence.

Next, $\mathcal{L}_{B_M}$ is for training ST-GCN1 and G-GRU1:

$$\mathcal{L}_{B_M} = \frac{1}{M}\sum \{\omega_1 ||\tilde{q}'_B - q'_B||_1 + \omega_2 ||\dot{\tilde{q}}'_B - \dot{q}'_B||_1$$
$$+ \omega_3 BL(\tilde{q}'_B, q'_B)\} + \omega_4 D_{KL}[z||\mathcal{N}(0,\mathbf{I})], \quad (6)$$

where $z$ is the latent variable, $\omega_4 = 1 - \omega_1 - \omega_2 - \omega_3$, $M$ is the total number of motions. $\tilde{q}'_B$ and $q'_B$ are the predicted and the ground-truth B motion. $\omega_1 = 0.75$, $\omega_2 = 0.1$ and $\omega_3 = 0.05$. $||\cdot||_1$ is the $l_1$ norm and $p(z|c) \sim \mathcal{N}(0,\mathbf{I})$. $BL(\tilde{q}'_B, q'_B)$ is the bone-length loss between $\tilde{q}'_B$ and $q'_B$:

$$BL(\tilde{q}, q) = \sum_t ||bone\_len(\tilde{q}^t) - bone\_len(q^t)||^2_2, \quad (7)$$

where $bone\_len$ computes the bone lengths of frame $t$ of $\tilde{q}$ and $q$. Note we minimize the difference between the ground-truth and prediction on the *zero-order* and *first-order* derivative in Eq. 6.

Summarily for $\mathcal{L}_{A_M}$:

$$\mathcal{L}_{A_M} = \frac{1}{M}\sum [\omega_1 ||\tilde{q}'_A - q'_A||_1 + \omega_2 ||\dot{\tilde{q}}'_A - \dot{q}'_A||_1$$
$$+ \omega_3 BL(\tilde{q}'_A, q'_A)] + \omega_4 D_{KL}[z||\mathcal{N}(0,\mathbf{I})], \quad (8)$$

where $z$ is the latent variable. $\omega_4 = 1 - \omega_1 - \omega_2 - \omega_3$, $M$ is the total number of motions. $\tilde{q}'_A$ and $q'_A$ are the predicted and the ground-truth B motion. $\omega_4 = 1 - \omega_1 - \omega_2 - \omega_3$, and $\omega_1 = 0.75$, $\omega_2 = 0.1$ and $\omega_3 = 0.05$. $BL(\tilde{q}'_A, q'_A)$ is the same bone length loss as in Eq. 7.

## 4. A New Interaction Dataset

To our best knowledge, there are few public datasets focusing on close and continuous interactions except [12]. To construct our dataset, we first obtain base motions and augment them. The base motion details are shown in the SM. We obtain "Judo". From CMU [1], we choose "Face-to-back", "Turn-around" and "Hold-body". From ExPI [12], we choose "Around-the-back", "Back-flip", "Big-ben", "Noser" and "Chandelle". These interactions are sufficiently complex to fully evaluate the robustness and generalizability of our model. They show the need for automated motion retargeting/generation as it requires hiring professional actors. Also, these motions contain rich and sustained contacts and close and continuous interactions, where single-body motion retargeting methods can easily lead to breach of contact and severe body penetrations.

After obtaining the base motions, a number of variations of each motion are collected to form a dataset. Our method is independent of how the variations are obtained. One may consider motion capture with actors of different body sizes, or manual keyframing with different characters. We employ a semi-automated approach. We manually change the skeleton to generate variations, after which we adapt an iterative and interactive optimization approach called InteractionMesh [16] to generate new motions based on the changed skeletons. This allows us to precisely control the bone sizes for rigorous and consistent evaluation.

For each base motion, we vary the bones by scales within [0.75, 1.25] with a 0.05 spacing, where the original skeleton is used as the scale-1 template skeleton. This spans the +-25% range of the original skeleton, covering most of the population. The process is semi-automatic, involving the use of an optimisation engine to carefully retarget an interaction to different body sizes, with manual adjustment of constraint weights and inspection of results. Synthesizing a few seconds of interaction generally requires around 2 minutes of computation. This is done multiple times for one variation of a base motion, due to the need for manual weighting tuning.

## 5. Experiments

### 5.1. Tasks, Metrics and Generalization Settings

**Tasks**. Since our model can generate motions with or without user input to specify a skeleton variation, we test different model variants for motion augmentation. Specifically, we evaluate our model on motion augmentation via retargeting and generation. If $B_s$ is given, we refer to the task as *retargeting* where we only use G-GRU1 and G-GRU2 for inference; if $B_s$ is not given, we use the full model (MLP2+G-GRU1+G-GRU2) and refer to it as *generation*.

**Metrics**. We employ four metrics for evaluation: joint position reconstruction error ($E_r$), bone-length error ($E_b$), Fréchet Inception Distance (FID), and joint-pair distance error (JPD). $E_r$, $E_b$ and JPD are based on $l_2$ distance. FID is used to compare the distributional difference between the generated motions and the data. JPD measures the key joint-pair distance error. The key joint pairs are the body parts in continuous contact. It is to investigate the key spatial relations between joint pairs in different motions (Judo: A's right hand to B's spine; Face-to-back: A's left hand to B's right hand; Turn-around: A's left hand to B's right hand; Hold-Body: A's right hand to B's spine; Around-the-back: A's left hand to B's right hand; Back-flip: A's left hand to B's right hand; Big-ben: A's right hand to B's right hip; Noser: A's right hand to B's right hip; Chandelle: A's right hand to B's right hip). All results reported are per joint results averaged over A and B.

**Generalization Settings**. Our dataset has two different

| Base Motion | M1 | M2 | M3 | M4 | M5 | M6 | M7 | M8 | M9 | Total |
|---|---|---|---|---|---|---|---|---|---|---|
| Original frames | 91 | 536 | 561 | 488 | 294 | 248 | 238 | 518 | 345 | 3,319 |
| Augmented motion | 160 | 119 | 119 | 119 | 90 | 90 | 90 | 90 | 90 | 967 |
| Augmented frames | 14,560 | 63,784 | 66,759 | 58,072 | 26,460 | 22,320 | 21,420 | 46,620 | 31,050 | 351,045 |

Table 1. M1: Judo, M2 Face-to-back, M3 Turn-around, M4: Hold-body, M5 Around-the-back, M6 Back-flip, M7 Big-ben, M8 Noser, M9 Chandelle. More details are in the SM.

skeletal topologies shown in the SM. Therefore, we divide them into two datasets: D1 (M1-4) and D2 (M5-M9) and conduct experiments on them separately. We employ four different settings to evaluate our model: *random*, *cross-scale*, *cross-interaction* and *cross-scale-interaction*:

1. *Random* means a random split on the data for training and testing where we keep 20% data for testing.
2. *Cross-scale* means we train on moderate bone scales but predict on larger skeleton variations. Our training data is within the scale [0.95, 1.05] and our testing data is both much smaller [0.75, 0.85] and larger [1.15, 1.25]. Note the testing varies up to +/- 25% of the bone lengths covering a wide range of bodies.
3. *Cross-interaction* is splitting the data by interaction types, e.g. training on Judo and tested dancing. When we choose one or several interactions for testing, the other interactions are used for training the model. Specifically, in D1, we split the data into two sets: M1-M2 and M3-M4; in D2, we split them into two sets: M5-M7 and M8-M9. In both, when one group is used for training, the other is used for testing.
4. *Cross-scale-interaction* is both cross-scale and cross-interaction, which is the hardest setting. This means that the scale [0.95, 1.05] of some interactions are used for training, and the scale [0.75, 0.85] and [1.15, 1.25] in the other interactions are for testing. For instance, in D1, when the scale [0.95, 1.05] of M1-M2 is used for training, the scale [0.75, 0.85] and [1.15, 1.25] in M3-M4 are for testing.

## 5.2. Evaluation

### 5.2.1 Retargeting and Generation

We present the main results here and refer the readers to the SM for more results and details.

We first show quantitative evaluation in Tab. 2. Across the two tasks, generation is harder than retargeting, as the bone scales are not given in generation. Naturally, the bone length error $E_b$ is almost always slightly worse than Retargeting and so is JPD. But even the worst case is 330% in $E_b$ and 206.89% worse in JPD which suggests the model generalizability on unseen scales and interactions in general is strong. We show visual results in Fig. 4 and the video. Together with the scaled skeleton, the poses are automatically adapted on both characters to keep the geometric relations

|  | | $E_r$ | $E_b$ | **JPD** | $FID$ | $E_b$ | **JPD** |
|---|---|---|---|---|---|---|---|
| M1 | Random | 1.069 | 0.171 | 3.008 | 2.934 | 0.18 | 3.421 |
|  | Cross-scale | 2.017 | 0.304 | 4.248 | 3.973 | 0.354 | 4.304 |
|  | Cross-interaction | 2.843 | 0.476 | 4.443 | 4.071 | 0.492 | 4.903 |
|  | Cross-scale-interaction | 3.021 | 0.679 | 4.754 | 4.369 | 0.753 | 5.067 |
| M2 | Random | 0.067 | 0.004 | 0.104 | 1.719 | 0.005 | 0.101 |
|  | Cross-scale | 0.344 | 0.018 | 0.241 | 2.364 | 0.023 | 0.645 |
|  | Cross-interaction | 0.671 | 0.087 | 0.625 | 3.077 | 0.097 | 1.004 |
|  | Cross-scale-interaction | 1.051 | 0.131 | 0.845 | 3.256 | 0.143 | 1.317 |
| M3 | Random | 1.076 | 0.02 | 2.274 | 5.573 | 0.03 | 2.134 |
|  | Cross-scale | 1.563 | 0.066 | 2.948 | 6.556 | 0.094 | 2.872 |
|  | Cross-interaction | 1.644 | 0.089 | 3.147 | 6.712 | 0.127 | 3.095 |
|  | Cross-scale-interaction | 1.928 | 0.13 | 3.493 | 6.863 | 0.153 | 3.317 |
| M4 | Random | 0.191 | 0.017 | 0.264 | 1.579 | 0.03 | 0.297 |
|  | Cross-scale | 0.471 | 0.079 | 0.418 | 2.148 | 0.087 | 1.071 |
|  | Cross-interaction | 0.617 | 0.104 | 0.589 | 2.648 | 0.111 | 1.347 |
|  | Cross-scale-interaction | 0.897 | 0.112 | 0.624 | 3.094 | 0.129 | 1.915 |
| M5 | Random | 1.975 | 0.003 | 0.398 | 0.69 | 0.01 | 0.604 |
|  | Cross-scale | 2.674 | 0.016 | 0.837 | 1.283 | 0.031 | 1.157 |
|  | Cross-interaction | 3.067 | 0.034 | 1.672 | 1.431 | 0.05 | 1.894 |
|  | Cross-scale-interaction | 3.864 | 0.067 | 2.268 | 1.897 | 0.094 | 3.068 |
| M6 | Random | 1.878 | 0.008 | 0.448 | 0.688 | 0.013 | 0.624 |
|  | Cross-scale | 3.615 | 0.022 | 0.997 | 1.22 | 0.028 | 1.273 |
|  | Cross-interaction | 4.013 | 0.031 | 1.923 | 1.523 | 0.039 | 2.024 |
|  | Cross-scale-interaction | 4.876 | 0.076 | 2.641 | 1.667 | 0.083 | 3.264 |
| M7 | Random | 2.746 | 0.006 | 0.495 | 0.645 | 0.015 | 0.702 |
|  | Cross-scale | 5.204 | 0.017 | 1.163 | 1.153 | 0.03 | 2.14 |
|  | Cross-interaction | 5.648 | 0.029 | 2.32 | 1.492 | 0.042 | 2.32 |
|  | Cross-scale-interaction | 5.757 | 0.066 | 2.759 | 1.475 | 0.069 | 3.762 |
| M8 | Random | 2.272 | 0.006 | 0.402 | 0.676 | 0.012 | 0.634 |
|  | Cross-scale | 3.124 | 0.021 | 0.964 | 1.349 | 0.038 | 1.374 |
|  | Cross-interaction | 3.389 | 0.04 | 1.534 | 1.671 | 0.057 | 1.862 |
|  | Cross-scale-interaction | 3.971 | 0.103 | 2.341 | 2.965 | 0.103 | 2.675 |
| M9 | Random | 2.234 | 0.005 | 0.403 | 0.634 | 0.009 | 0.561 |
|  | Cross-scale | 2.935 | 0.01 | 0.934 | 1.412 | 0.043 | 1.259 |
|  | Cross-interaction | 3.256 | 0.023 | 1.674 | 1.842 | 0.051 | 1.903 |
|  | Cross-scale-interaction | 3.623 | 0.064 | 2.842 | 2.854 | 0.114 | 2.971 |

Table 2. Retargeting (left) and Generation (right). Here is the result of D1 (M1-4) and D2 (M5-9).

of the interaction.

In terms of generation settings, the overall difficulty should be Cross-scale-interaction > Cross-interaction > Cross-scale > Random, as more and more information is included in the training data from Cross-scale-interaction to Random. The metrics in Tab. 2 are consistent with this expectation. Cross-scale-interaction is the most challenging task which is testing the model on both unseen bone sizes and interactions simultaneously. Its metrics are worse than the other three in general as expected. Despite the worse results, the visual results of cross-scale-interaction are of good quality. We show one example (with the worst metrics) in Fig. 5 in comparison with ground-truth.

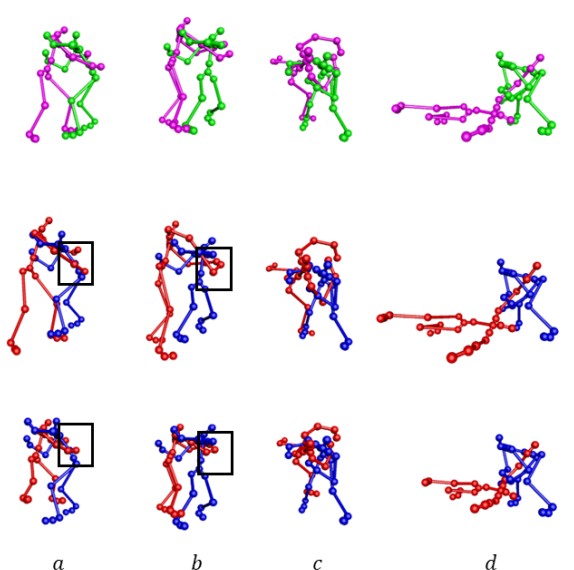

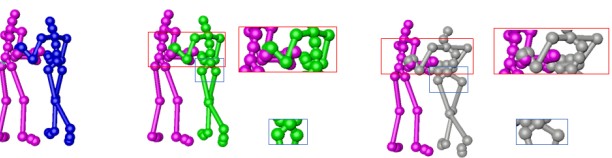

|  | Hold-Body | | | Judo | | |
|---|---|---|---|---|---|---|
|  | Our method | [35] | [11] | Our method | [35] | [11] |
| FID | **0.412** | 2.257 | 40.351 | **0.267** | 1.998 | 28.459 |
| Eb | **0.002** | 0.541 | 0.389 | **0.118** | 0.334 | 0.311 |
| JPD | **0.168** | 1.463 | 4.903 | **3.401** | 4.532 | 5.648 |

Table 3. Results at Scale 1.25, averaged over 10 randomly generated motions.

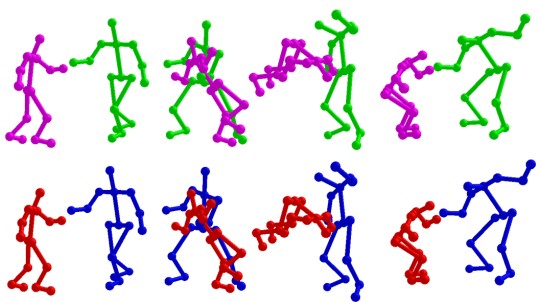

Figure 6. Scale 1.25 comparisoin. Left: ground-truth, mid: ours, right: [35]. [35] generates unnatural poses and break contact (enlarged parts). Zoom-in for better visualization.

*a      b      c      d*

Figure 4. In the original Judo motion (top), the red character is augmented for a bigger body (middle) and a smaller body (bottom), while retaining the key features of the interaction semantics. The black boxes in column **a** highlight how the "Judo holding" semantics, i.e., the red character holding the blue one, are adapted. The black boxes in column **b** show a similar example.

## 5.3. Comparison

To our best knowledge, it is new for deep learning to be employed for interaction augmentation with varying body sizes. So there is no similar research. Therefore, we adapt two single-body methods ([11, 35]) which provide conditioned generation and are the only methods we know that could potentially be adapted for handling varying bone lengths, i.e. we train the model by labelling different scales as different conditions and train the model on scale [0.75, 1.25]. More specifically, both models require action type (i.e. a class label) as input, so we label data at different scales as different classes. Note [35] and [11] cannot generate motions for unseen action types, which means they cannot predict on unseen scales like our method.

We show the metrics in Tab. 3. After trying our best to train [11], it still generates jittering motions. It can preserve the bone-length better than [35] but its FID and JPD are much worse. [35] generate better results but it is still much worse than our method. We show one example of Hold-Body in Fig. 6 in comparison with [35]. Overall, single-body methods even when adapted cannot easily generate interactions.

We also compare with InteractionMesh [16]. Since our ground-truth is from InteractionMesh, comparisons on the aforementioned evaluation metrics would be meaningless. Instead, we compare the speed and motion quality on unseen extreme scales. The inference time of our model is 0.323 seconds, while InteractionMesh needs ∼120 seconds on average per optimization, plus the time needed for manual tuning of the weighting. Admittedly, our model needs overheads for training. However, once trained, it is very fast and can be used for interactive applications. Further, InteractionMesh needs to do optimization for every given $B_s$, while our model is trained once then does inference for any $B_s$. Last but not least, InteractionMesh sometimes fails to

Figure 5. Comparison between ground-truth (top) and cross-scale-interaction (bottom). The skeleton of the red character is changed. Both of them are Back-flip on scale 0.85.

### 5.2.2 Extrapolating to Large Unseen Scales

We predict larger scales. The scales are beyond our dataset (including the testing data). We show one example of Turn-around on 0.65 and 1.3 in the SM , which shows that our model can extrapolate to larger skeletal variations when trained only using data on scales [0.95, 1.05]. More examples can be found in the video. Although larger scale variations e.g. 0.5 and 1.5 might lead to unnatural motions, the SM already demonstrate the generalizability of our model.

| Predict(sec) | | 0.2 | 0.4 | 0.6 | 0.8 | 1.0 |
|---|---|---|---|---|---|---|
| M5 | JME | **0.234**/0.449 | **0.427**/0.771 | **0.593**/1.073 | **0.722**/1.365 | **0.848**/1.594 |
| | AME | **0.417**/0.605 | **0.750**/1.100 | **1.036**/1.499 | **1.250** /1.877 | **1.474**/2.176 |
| M6 | JME | **0.520**/0.552 | **0.848**/0.874 | **1.098**/1.187 | **1.485**/1.533 | **1.670**/1.799 |
| | AME | **0.671**/0.682 | 1.170/**1.168** | **1.530**/1.579 | **1.958**/1.968 | **2.253**/2.326 |
| M7 | JME | **0.538**/0.565 | 0.971/**0.959** | **1.298**/1.302 | 1.720/**1.708** | 1.926/**1.848** |
| | AME | **0.708**/0.727 | **1.334**/1.367 | **1.809**/1.823 | 2.319/**2.290** | 2.608/**2.573** |
| M8 | JME | **0.507**/0.562 | **0.927**/0.985 | **1.137**/1.284 | **1.648**/1.692 | **1.886**/2.013 |
| | AME | **0.673**/0.695 | **1.315**/1.330 | **1.796**/1.830 | **1.908**/1.968 | **2.353**/2.483 |
| M9 | JME | **0.505**/0.590 | **0.834**/0.920 | **1.263**/1.312 | **1.567**/1.725 | **1.904**/2.201 |
| | AME | **0.721**/0.723 | **1.469**/1.634 | **1.848**/1.923 | **2.031**/2.224 | **2.415**/2.657 |
| M5 | JME | **0.278**/0.507 | **0.444**/0.767 | **0.652**/1.122 | **0.763**/1.299 | **0.867**/1.641 |
| | AME | **0.467**/0.668 | **0.748**/1.094 | **1.085**/1.603 | **1.345** /1.894 | **1.551**/2.230 |
| M6 | JME | **0.538**/0.548 | **0.856**/0.880 | **1.096**/1.180 | **1.488**/1.586 | **1.622**/1.793 |
| | AME | **0.683**/0.690 | **1.194**/1.196 | **1.528**/1.566 | **1.960**/1.973 | **2.256**/2.335 |
| M7 | JME | 0.584/**0.579** | **1.023**/1.049 | 1.322/**1.315** | **1.645**/1.648 | **1.937**/1.940 |
| | AME | **0.723**/0.746 | **1.466**/1.489 | **1.896**/1.900 | 2.391/**2.379** | **2.608**/2.612 |
| M8 | JME | **0.597**/0.605 | **1.036**/1.068 | **1.204**/1.315 | **1.701**/1.767 | **1.892**/2.148 |
| | AME | **0.710**/0.748 | 1.348/**1.347** | **1.808**/1.810 | **2.064**/2.101 | **2.332**/2.425 |
| M9 | JME | **0.524**/0.528 | **0.862**/0.892 | **1.378**/1.392 | **1.674**/1.702 | **1.923**/2.046 |
| | AME | 0.718/**0.713** | **1.486**/1.497 | **1.867**/1.901 | **2.067**/2.209 | **2.523**/2.672 |

Table 4. Motion prediction of [12] (top) and [56] (bottom) in JME (joint mean error) and AME (aligned mean error) from D2 (M5-9). In each test, xx/xx is with/without data augmentation.

| Settings/Classifiers | HD-GCN [23] | STGAT [19] | TCA-GCN [55] |
|---|---|---|---|
| 80/10/10 | **94.80**/94.36 | **94.27**/94.10 | **94.68**/94.62 |
| 50/20/30 | **93.92**/92.65 | **93.66**/92.40 | **93.27**/91.38 |

Table 5. Activity recognition accuracy on 3 different classifiers from ExPI [12]. In each test, xx/xx is with/without data augmentation.

| Methods/Classifiers | HD-GCN [23] | STGAT [19] | TCA-GCN [55] |
|---|---|---|---|
| **ACTOR**[35] | 97.68 | 98.03 | 97.22 |
| **Action2motion**[11] | 97.43 | 96.90 | 96.45 |
| Our method | **98.64** | **98.53** | **97.93** |

Table 6. Activity recognition accuracy on 3 different methods from D2 (M5-9). Training on the ground-truth and testing on generated 200 motions.

converge due to its optimization set up, resulting in either numerical explosion or very unnatural motions (see video). This requires careful manual tuning. Comparatively, our model does not need manual intervention.

## 5.4. Downstream Tasks

Motion augmentation can benefit various downstream tasks. Here we show two downstream tasks: motion prediction and activity recognition. In motion prediction we train two models [56] and [12] on the ExPI dataset [12] with/without our data augmentation, following their settings. The testing protocols and evaluation metrics follow [12]. The results are shown in Tab. 4, where 90 of 100 metrics are improved by our augmentation, with a maximum 47.88% improvement on JME (M5-AB-0.2sec) and a maximum 47.74% improvement on AME (M5-AB-0.6sec).

In activity recognition, we train three latest activity classifiers HD-GCN [23], STGAT [19] and TCA-GCN [55] on ExPI with/without data augmentation, following two data splits: 80/10/10 and 50/20/30 split on training/validation/testing data. The results are shown in Tab. 5. The data augmentation improves the accuracy across all models and all split settings. As the training data is reduced from 80% to 50%, the results with data augmentation have a small deterioration (less than 1.49%). Without data augmentation, it quickly drops by as much as 3.42%.

We further show the quality of the augmented motions via a trained classifier. If a trained classifier can correctly recognize the generated motions, then it suggests the generated features have similar features to the original data. We train the aforementioned classifiers on the original ExPI data and use the generated motions as testing data. Tab. 6 shows the action recognition result. Our method outperforms the other two methods in all three action recognition classifiers, which shows that our generated data has more similar features to the ground-truth. Given close interaction data is new [12] and its limited variety and amounts, our method provide an efficient way of augmenting such data for activity recognition.

## 5.5. Alternative Architectures

Our model combines existing network components in a novel way for interaction augmentation, so a natural question is if there are other better alternative architectures. We test several alternative network architectures inspired by existing research. The selection criteria is they need to be data efficient for learning, so we exclude some data-demanding architectures such as Transformers or Diffusion models. The details and results are shown in the SM, but overall our model outperforms the alternative architectures.

## 6. Conclusion, Limitations & Discussion

To our best knowledge, our research is the very first deep learning model for interaction augmentation. It has high accuracy in generating desired skeletal changes, great flexibility in generating diversified motions, strong generalizability to unseen and large skeletal scales, and benefits to multiple downstream tasks. One limitation is that we need some data samples to start and require the same skeletal topology to do cross-motion motion augmentation. However, considering the difficulties of interaction motion capture, our method provides a new and fast way of iteratively augmenting a single captured motion then learning to generate infinite number of variations. Next, although we use InteractionMesh to generate training data, our method can easily incorporate other data sources such as captured motions from different subjects as well as manually created motions by animators. Given the small number of motions needed by our method, this is still a fast pipeline to acquire a large number of interactions with varying body sizes.

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
