# OpenReview forum: "Two-Person Interaction Augmentation with Skeleton Priors"
_thecvf.com/CVPR/2024/Workshop/HuMoGen — CVPR 2024 Workshop HuMoGen Submission_

### Official Review · Reviewer_MZfD · 2024-03-31

**Rating:** 4
**Confidence:** 4

**Review:**

# Summary
This paper focuses on close and continuous human interaction with rich contacts. A method is proposed to learn two-body skeletal interaction to generate contact-rich interactions with varying body sizes and proportions: predicting the joint relation distributions based on a given skeleton (specifically, bone length vector). Furthermore, the method allows for learning from a relatively small amount of data and generalize to drastically different skeleton sizes.  To train the model, the authors also introduce a new dataset augmented from single interaction examples with different body sizes and proportions using the existing method InteractionMesh.

# Exposition
“To our best knowledge, there is no deep-learning method for complex two-character interactions. ”-> This sentence is too big; it is better to revise it.

The resolution of Fig1 should be enhanced.

L256 - 257 “shown in Fig. 1 Compared xxx” -> shown in Fig. 1. Compared

“SM” is used for the short of Supplement Material… Consider using “The Supplementary” or “Appendix”...

# References

Generally OK.

Some recent related references are missing:
- For single-person human motion generation:

Generating diverse and natural 3d human motions from text. CVPR22.

T2M-GPT: Generating Human Motion from Textual Descriptions with Discrete Representations, CVPR 2023.

…

- For two-person interactions:

Strategies for Physically Simulated Characters Performing Two-player Competitive Sports, ACM Transactions on Graphics (SIGGRAPH 2021).
- For skeleton-agnostic retargeting:

SAME: Skeleton-Agnostic Motion Embedding, SIGGRAPH Asia 2023.

# Technical Correctness

Yes. The whole pipeline makes sense to me.
The encoder part uses Spatio-Temporal Graph Convolution Networks to capture spatial and temporal information about the skeleton, while the decoder uses a gated Recurrent Unit to reconstruct the body motion. The dataset is constructed from a single-person motion dataset by employing the off-the-shelf model InteractionMesh.  The innovative aspect of the proposed model is somewhat constrained. but it can be taken as a step for generating two-person motions considering varying body sizes (skeletal structure). The evaluation validates the method to some extent.
I highly suggest the authors release their codes for reproduction purposes.


# Overall Recommendation

The whole pipeline is reasonable, though the novelty is still a little limited. The tested dataset scale is not that large.
But generally, this is a well-made paper for publishing at the workshop.

---

### Official Review · Reviewer_8SvY · 2024-03-31
**The authors present a deep learning based approach for two person skeletal interaction motion augmentation. The method is technically sound and has potential to benefit motion generation with multiple skeleton sizes without capturing such data..**

**Rating:** 4
**Confidence:** 4

**Review:**

The paper proposes a learning based method that can be used to generate interaction between different skeleton sizes without the need to capture data with such skeletons. The method is able to learn the joint relation distribution between two characters based on a given skeleton and retarget the relation to another skeleton of different size.
Strengths:
1.  The idea is novel and interesting to explore since it benefits from not having to capture more motions with different sizes of skeletons. The method is technically sound.
2. The paper is clearly written and easy to understand.
3. The paper provides adequate baselines for the tasks of motion prediction and activity recognition. However, [12] could also be used as another baseline for the motion prediction since it involves two person interaction.

Weakness/Questions:
1. The difference between the ground-truth and the generated motions are not identifiable.
2. Even though the two person interaction is being modeled with STGCN, how does the method make sure that the contact points are intact? E.g. wrapping one's arms around the other's body, how is it ensured that the contact points between the arms are correct when the skeleton sizes are changed.
3. The related work section can benefit from adding some recent deep learning based two-person interaction generation methods such as  InterGen[*1], ReMoS[*2], ReGenNet[*3]. The first paper also propose two-person datasets, which might be worth looking into.

[*1] Liang, Han, et al. "Intergen: Diffusion-based multi-human motion generation under complex interactions." International Journal of Computer Vision (2024): 1-21.
[*2] Ghosh, Anindita, et al. "ReMoS: Reactive 3D Motion Synthesis for Two-Person Interactions." arXiv preprint arXiv:2311.17057 (2023).
[*3] Xu, Liang, et al. "ReGenNet: Towards Human Action-Reaction Synthesis." arXiv preprint arXiv:2403.11882 (2024).

---

### Meta-Review · Area_Chair_xGnU · 2024-04-05

**Recommendation:** Accept

**Metareview:**

The paper received two weak accepts. The paper contributes augmentations of the ExPI dataset, the algorithm design is reasonably novel to clear the bar of a workshop paper. The results presented in the supplementary video also present a convincing picture. Therefore, the AC recommends to accept this paper. There were concerns regarding model overfitting as test results look similar to the ground-truth. The authors are encouraged to address this concern in their revised camera ready.

---

### Decision · Program_Chairs · 2024-04-06

**Decision:**

Accept

**Comment:**

The paper will be published as part of the official CVPR workshop proceedings upon submission of the camera-ready version.